# COVID-19 Vaccine Acceptance in Azuay Province, Ecuador: A Cross-Sectional Online Survey

**DOI:** 10.3390/vaccines9060678

**Published:** 2021-06-21

**Authors:** Julio Jaramillo-Monge, Michael Obimpeh, Bernardo Vega, David Acurio, Annelies Boven, Veronique Verhoeven, Robert Colebunders

**Affiliations:** 1Faculty of Health Science, Universidad de Cuenca, Cuenca 010203, Ecuador; julio.jaramillo@ucuenca.edu.ec (J.J.-M.); bernardo.vegac@ucuenca.edu.ec (B.V.); david.acurio@ucuenca.edu.ec (D.A.); 2Family Medicine and Population Health, University of Antwerp, 2610 Antwerp, Belgium; Michael.Obimpeh@student.uantwerpen.be (M.O.); Annelies.Boven@student.uantwerpen.be (A.B.); veronique.verhoeven@uantwerpen.be (V.V.)

**Keywords:** COVID-9, Ecuador, vaccine acceptance, hesitancy

## Abstract

We investigated the COVID-19 vaccination acceptance level in Azuay province, Ecuador through an online survey from 12th to 26th February (before the start of the COVID-19 vaccination campaign in Ecuador). Overall, 1219 respondents participated in the survey. The mean age was 32 ± 13 years; 693 participants (57%) were female. In total, 1109 (91%) of the participants indicated they were willing to be vaccinated with a COVID-19 vaccine, if the vaccine is at least 95% effective; 835 (68.5%) if it is 90% effective and 493 (40.5%) if it is 70% effective; 676 (55.5%) participants indicated they feared side effects and 237 (19.4%) thought the vaccine was not effective. Older age, having had a postgraduate education, a history of a negative COVID-19 test, a high level of worry of contracting COVID-19, believing that COVID-19 infection can be prevented with a vaccine and understanding there is currently an effective vaccine against COVID-19 were associated with higher vaccination acceptance. A vaccination education campaign will be needed to increase the knowledge of Ecuadorians about the COVID-19 vaccine and to increase their trust in the vaccine. People with a lower education level and living in rural areas may need to be targeted during such a campaign.

## 1. Introduction

COVID-19 is a severe respiratory disease caused by the virus SARS-CoV-2. By 18 April 2021, it was responsible for more than 3 million deaths and more than 140 million infections worldwide [1]. Almost 350,000 cases and more than 17,000 deaths occurred in Ecuador; almost 20,000 cases, including 15,000 in the city of Cuenca, and more than 350 deaths occurred in Ecuador’s province of Azuay [2].

None of the currently available medicines can cure COVID-19, which is why prevention remains the main strategy to avoid becoming infected and developing critical illness, which eventually leads to death. This strategy is based on mobilising everyone to take hygiene and physical distancing measures, appropriate quarantine and isolation of (possibly) infected individuals, community restrictions, providing appropriate clinical care and finally developing safe and effective vaccines [3].

The vast majority of the vaccines against SARS-CoV-2 target either its full-length spike (S) protein or only this protein’s receptor-binding domain [4]. For entry of the virus into the host cell, this domain binds to angiotensin-converting enzyme 2, a human membrane receptor that normally induces an anti-inflammatory response when it is not bound by this virus [5].

As new technologies have allowed for a reduced vaccine development time, 13 vaccines have already been licensed for general use by at least one country [6].

Despite this rapid development of vaccines, the waiting period before receiving a vaccine is much higher for people in low-income countries than for people in higher-income countries. This is due to unequal vaccine distribution across the world. At least 70% (4.2 billion) of the doses of COVID-19 vaccines produced in 2021 have been secured by high-income countries who represent only 16% of the world population [7]. Moreover, more than 84% of the vaccines have already been administered to people in high- and upper-middle-income countries, whereas only 0.1% have been administered to people in low-income countries [8].

The COVAX Facility aims to distribute the vaccines more equally across the world by sharing the vaccines with lower-income countries. At present, COVAX has delivered 38 million vaccines to 98 countries [9]. Thus far, 84,000 doses have been delivered to Ecuador, and 6,802,000 doses are expected to be delivered from April 2021 on [10]. Ecuador has additionally received both Pfizer (341,710) and Sinovac (1,000,000) doses. Cumulatively, Ecuador has been allocated more Pfizer (5,800,000) compared to Sinovac (2,000,000) and AstraZeneca (5,000,000) doses [10]. Ecuador’s mass vaccination programme started between the end of March and beginning of April 2021 and has ensured the administration of 250,631 first doses and 112,624 second doses [11].

Besides the availability of the vaccines, other conditions are required to successfully vaccinate Ecuador’s entire population, such as a consensus on the order in which population groups need to be vaccinated and a plan to address the worries and preoccupations of the public regarding the safety of the vaccination [12]. The latter is very important for public acceptance of the COVID-19 vaccines before the public is vaccinated [13].

The COVID-19 vaccination acceptance rate in China was found to be above 90% in one study [14], and 87% in another [15]. High acceptance rates were also observed in Brazil, South Africa, South Korea, Mexico and the United States, ranging from more than 75% in the United States to more than 85% in Brazil [15]. In April 2020, a survey organised in Ecuador demonstrated that among 1050 households, 97% would accept a COVID-19 vaccine and 85% would be willing to pay for a vaccine [16]. In June 2020, a 19-country online survey was organised to assess the COVID-19 vaccine acceptance rate; of the 741 persons from Ecuador who participated in this survey, 72% answered that they would be willing to be vaccinated if the vaccine was proven to be safe and effective [15].

According to the World Health Organization (WHO), vaccine hesitancy, which is defined as a “delay in acceptance or refusal of vaccination despite availability of vaccination services” [17], is mostly influenced by three factors: (1) confidence, or trust in the safety and effectivity of the vaccine; (2) complacency, or perception that vaccination is valuable and necessary; and (3) the convenience of the vaccination, which includes the accessibility, affordability and availability of vaccination services [17].

Factors influencing the COVID-19 vaccination acceptance in China included both confidence-related factors, such as the effectiveness of the vaccine, the side effects, duration of protection and number of required vaccine doses, as well as convenience-related factors such as the number of acquaintances who had been vaccinated, access to the vaccine and the vaccination site and, to a lesser extent, the cost [18,19].

In a cross-sectional study that included 60 countries, 10 on each continent, another confidence-related factor that influenced the willingness to accept a COVID-19 vaccination was a high level of trust in the information provided by the government [20].

The effectiveness and safety of COVID-19 vaccines were associated with vaccine acceptance in the United States [21], but other factors, such as political preferences, age, gender and ethnicity, additionally influenced the vaccine acceptance [22].

In Brazil, Colombia and Guatemala, a major factor that influenced general vaccine acceptance was fear, especially fear of adverse events [23]. Other important factors in Latin America included a lack of information or distrust in the information about the vaccine (the confidence-related factors) [23,24], refusal based on cultural, religious or one’s own beliefs (the complacency-related factor) [23,24], inaccessibility of healthcare services, cost and economic issues (the convenience-related factors) [23,24] and socio-demographic factors like socio-economic and educational status [24]. The inaccessibility in this region was influenced by additional factors such as armed conflict, social instability or meteorological conditions that make the access to these services more difficult [23]. Previous studies have furthermore indicated that general vaccination acceptance is influenced by the involvement of healthcare workers in the Dominican Republic, or religious leaders in Guatemala, in supporting the vaccination campaign and by providing information about the vaccine [23].

The anti-vaccination movement is another group that has a great influence on the vaccine acceptance. This group tries to decrease people’s trust in their government and the vaccines, by spreading misinformation about the COVID-19 vaccines via social media and targeting people who are unsure about their willingness to be vaccinated [25].

In this study, conducted at the start of COVID-19 vaccinations in Ecuador, we aimed to collect information about the opinions of Ecuadorians from Azuay Province about COVID-19 vaccines, their willingness to be vaccinated and reasons for vaccine hesitancy.

## 2. Materials and Methods

By means of an online survey from 12th to 26th February, we tried to reach voluntary participants, aged at least 18 years old, and asked them to answer questions to the survey which was developed by the International Citizen Project COVID-19 (ICPCovid-19). This study was part of a network of online surveys organised by the International Citizen Project COVID-19 (ICPCovid; online platform available at: https://www.icpcovid.com/en/home (accessed on 04 February 2021), which uses web-based surveys to investigate the impact of COVID-19 and associated restrictions on residents of several low- and middle-income countries. The English version of the ICPCovid online questionnaire was translated into Spanish and adapted to the local context in Ecuador. We used a snowball approach by sharing the questionnaire through WhatsApp and Facebook Inc Menlo park, California, and Twitter Inc, San Francisco, and emails from the University of Cuenca, while asking participants to further distribute the questionnaire to their contacts.

Using the online tool, we asked participants for informed consent prior to data collection. We collected socio-demographic data, data on vaccine acceptability at different levels of effectiveness, perceptions of vaccination against COVID-19, knowledge about the vaccine and vaccination in the population and having been tested for COVID-19. All participants provided an e-consent. The protocol was approved by the ethics committee of the University of Cuenca (COBIAS).

### Data Processing and Analysis

Completed questionnaires were extracted from the secured server of the ICPCovid website, exported to a Microsoft Excel 2016 spreadsheet for cleaning and coding and subsequently transferred to R software for analysis.

Categorical variables were summarised as frequencies and proportions, continuous variables as mean and standard deviation (SD) or median and interquartile range (IQR). The association between dependent and independent variables was determined using adjusted odds ratios (ORs), with 95% confidence intervals (95% CIs). *p*-values < 0.05 were considered significant.

A multiple logistic regression analysis was used to investigate factors associated with COVID-19 vaccine acceptance. The associations between dependent and independent variables were determined using both crude odds ratios (CORs) and adjusted odds ratios (AORs), with 95% confidence intervals (95% CIs) and a *p*-value < 0.05 to determine the statistical significance level of the independent variables.

Bivariate regressions were carried out to identify potential variables associated with COVID-19 vaccine acceptance. The variables with a likelihood ratio *p*-value < 0.25 in bivariate regression were included in the multiple logistic regression analysis. The selected variables from the bivariate analysis were subjected to a backward stepwise selection process and a final best performing model with the smallest Akaike information criterion (AIC) was selected. Multi-collinearity was checked using variance inflation factors, hence the variables in the models did not depend on each other, and did not render the models inaccurate in our parameter estimations. The level of significance used was 5% and all tests were two sided.

## 3. Results

### 3.1. Participants’ Characteristics

Overall, 1219 respondents participated in the survey. The mean age was 32 ± 13 years; 693 participants (57%) were female; 300 (24.6%) participants had a secondary level of education, 549 (45%) had an undergraduate level of education and 370 (30.3%) had a postgraduate level of education (Table 1). One hundred and sixty-one (13.2%) reported to suffer from a chronic or underlying disease; 670 (55%) had been tested for COVID-19, of which 129 (19%) had a positive result.

Participants from urban areas were more likely to have a higher education level and to have a higher income. Participants from a rural compared to an urban area were less likely to believe that a COVID-19 infection can be prevented by a vaccine (58% versus 70.8%) (X^2^ = 13.047, df = 2, *p*-value = 0.0014), and that currently there is an effective vaccine (46.9% versus 59.3%) (X^2^ = 10.465, df = 2, *p*-value = 0.0053). A high proportion of both participants from rural and urban areas (85.5% versus 88.4%) believed that someone can be re-infected after recovering from a previous COVID-19 infection.

### 3.2. Vaccine Acceptance

In total, 1109 (91%) participants indicated that they were willing to be vaccinated with a COVID-19 vaccine, if the vaccine is at least 95% effective, 835 (68.5%) if it is 90% effective, 493 (40.5%) if it is 70% effective and 329 (27%) if the vaccine is 27% effective (Figure 1).

Six hundred and seventy-six (55.5%) participants indicated that they feared unforeseen side effects; 237 (19.4%) believed the vaccine was not effective (Figure 2).

### 3.3. Determinants of COVID-19 Vaccine Acceptance

In a multiple logistic regression, six variables were associated with COVID-19 vaccine acceptance (Table 2). Participants with a unit increase in age, participants with a postgraduate education, participants with a positive COVID-19 test, participants with increasing levels of worry and fear of contracting COVID-19, participants who believe COVID-19 infection can be prevented with a vaccine and participants who understand there is currently an effective vaccine against COVID-19 were more likely to accept a COVID-19 vaccine.

In a sensitivity analysis accounting for the effect of area of residence and schooling as clusters, we did not observe significant changes in the results of our multiple logistic regression analysis (details not shown).

## 4. Discussion

At the start of the COVID-19 vaccination campaign in Ecuador, we wanted to know the opinion of the residents in the province of Azuay about the efficacy of COVID-19 vaccines, their willingness to be vaccinated and reasons for COVID-19 vaccine hesitancy. An online survey was organised to rapidly collect useful information for planning the campaign.

Belief in the efficacy of COVID-19 vaccines was particularly low in rural areas, where only 58% of the participants believed that a COVID-19 infection can be prevented by a vaccine, and only 46% believed that there is currently an effective COVID-19 vaccine. In both rural and urban areas, a large proportion of participants (about 85%) were aware that someone can be re-infected after recovering from a previous COVID-19 infection.

Overall, 91% of the participants indicated they were willing to be vaccinated with a COVID-19 vaccine if the vaccine is at least 95% effective, 68.5% if it is at least 90% effective and 40.5% if it is at least 70% effective.

Approximately 55.5% of the participants indicated they feared unforeseen side effects. Even though the available COVID-19 vaccines have been declared to be safe, the long-term effects are not completely known. Therefore, a considerable number of participants expressed worry about the vaccine’s side effects. Furthermore, 19.4% of the participants thought the vaccine was not effective.

In addition, older age, a higher level of education, a history of a positive COVID-19 test, a high level of worry, a belief that COVID-19 infection can be prevented with a vaccine and a belief there is currently an effective vaccine against COVID-19 increased the probability of accepting a COVID-19 vaccination.

The COVID-19 vaccine acceptance rate in this study (91%) is quite consistent with the rate in Ecuador reported in April 2020 (97%) [16] but is almost 20% higher than the rate reported in June 2020 (72%) [15]. These variations might be caused by different time periods, and/or differences in study samples. These study samples might not be very comparable, since both previous studies included people from the entire population of Ecuador [15,16], whereas our study only included people from Azuay Province. Both our study and the survey organised in April 2020 included more people from urban regions and with a significantly higher level of education than Ecuador’s general population [16], while the study conducted in June 2020 included a more representative sample of Ecuador’s general population [15]. In a recent survey about COVID-19 acceptance with the same ICPCovid questionnaire conducted in several other low- and middle-income countries, the vaccine acceptance rate in Brazil was 94.2%, in Thailand 87.3% and in Malaysia 78.6%, but it was much lower in most African countries, for example, only 59.4% in the DRC for a vaccine that is 95% effective [26].

Another explanation for this difference in vaccine acceptance rates might be that both previous studies did not distinguish between different vaccine efficacy levels, which is an important factor as it is associated with different levels of vaccine acceptance.

To reach a sufficiently high vaccination coverage, vaccine acceptance must be increased. An efficient way to do this would be targeting subgroups with higher vaccine hesitancy levels, such as the groups identified in this study: younger people, people with a lower level of education and people not worried about infection. These people must be addressed with tailored information to increase their knowledge and to enable informed decision making.

Previous interventions have identified the following measures to be effective in increasing voluntary influenza vaccine uptake: addressing concerns and putting the benefits and risks into a non-persuasive perspective [27]. Furthermore, quick access to anti-vaccine messages on websites and social media significantly decreases vaccine acceptance [28], which is why these messages should be tackled using good communication and community engagement [12].

Engaging the community could be conducted by involving both healthcare workers and religious leaders in providing adequate information and in openly supporting vaccination, in order to increase vaccine acceptance [23]. Even though this increase is very important, other steps need to be taken as well to achieve good vaccine coverage. Improving access and convenience of access to the vaccine will positively influence vaccination uptake, and activation by means of reminders, personal invitations or information leaflets will increase uptake in people who are willing to have the vaccine [29].

## 5. Limitations of the Study

Several limitations of our study must be acknowledged. First, our respondents were not representative of the general population, since only literate persons with access to the internet were able to participate and were recruited by convenience and snowball sampling, which introduced a selection bias. This resulted in more participants from urban regions and with a higher level of education than the general population. Second, self-reports may be influenced by recall bias and social desirability bias. Third, we did not use a validated questionnaire that contained control questions. Our questionnaire additionally assessed most, but not all, factors influencing vaccine hesitancy, because it did not consider all factors of the recently introduced 5C scale model [30]. This model has been proposed, instead of the 3C scale model (confidence, complacency and convenience), to monitor vaccine hesitancy by assessing acceptance, access, affordability, awareness and activation [30]. Finally, our study was only a cross-sectional study conducted over a short time. Opinions and attitudes concerning COVID-19 vaccination may change rapidly if side effects of vaccines appear and because of social media influence. It is important to note that our survey was performed before the AstraZeneca and Johnson & Johnson side effects of rare blood clots were reported [31,32]. According to a recent declaration of the Minister of Health of Ecuador, around 30% of persons invited to be vaccinated do not show up for vaccination. Possible explanations could be a logistical or mobility problem of the elderly and a lack of trust in the AstraZeneca and Sinovac vaccine. A follow-up survey should be considered to monitor the effect of the vaccines’ reported side effects and information about new COVID-19 strains that may interfere with the efficacy of the vaccines.

## 6. Conclusions

In our study population, the acceptability of the COVID-19 vaccine mainly depended on the level of effectiveness and safety of the vaccines. Like studies conducted worldwide, the fear of side effects was high. Vaccine hesitancy was particularly high in younger people and people with a lower education level. As younger people are less at risk for developing severe COVID-19, vaccination is less important for them. However, they are potential transmitters of the virus. Therefore, as soon as sufficient vaccines are available, it will be important to convince them to agree to be vaccinated. Recent studies suggest that vaccination decreases infectiousness, but people should know that a vaccinated person can still transmit the virus [33]. More effort is needed to increase the knowledge of the population about the benefit of COVID-19 vaccination for individuals and society, but also about the safety of the vaccines. Increasing knowledge will increase people’s trust in the vaccine. During a vaccination education campaign, special attention should go towards people with a lower education level and people living in rural areas. Finally, follow up surveys to monitor vaccine hesitancy in random samples of populations should be considered using a validated scale for measuring vaccine hesitancy.

## Figures and Tables

**Figure 1 vaccines-09-00678-f001:**
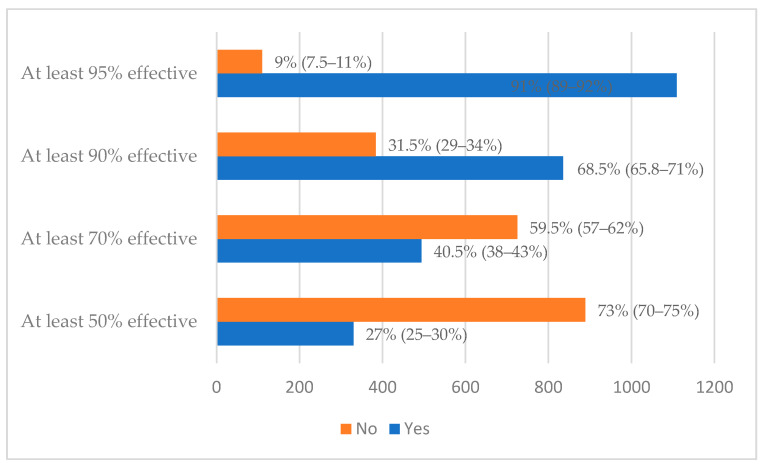
Acceptance rate of COVID-19 vaccine with confidence intervals.

**Figure 2 vaccines-09-00678-f002:**
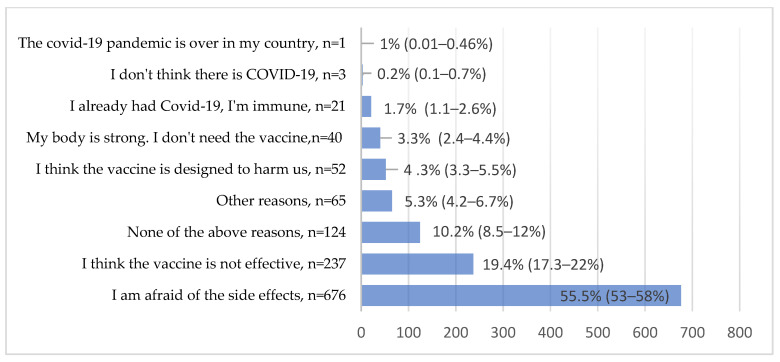
Reasons for COVID-19 vaccine hesitancy with confidence intervals.

**Table 1 vaccines-09-00678-t001:** Participants’ demographic characteristics according to place of residence and opinions about COVID-19 vaccines.

Variables	Response	Place of Residence		
		Rural	Suburban	Urban
Sex	Male, *n* (%)	84 (40.6%)	85 (53.8%)	357 (41.8%)
	Female, *n* (%)	123 (59.4%)	73 (46.2%)	497 (58.2%)
Age	Mean (SD)	26.68 (10.04)	27.8 (10.6)	33.5 (13.3)
Education	Secondary, *n* (%)	73 (35.3%)	53 (33.5%)	174 (20.4%)
	Undergraduate degree, *n* (%)	106 (51.2%)	67 (42.4%)	376 (44%)
	Postgraduate degree, *n* (%)	28 (13.5%)	38 (24.1%)	304 (35.6%)
Current socio-economic situation	Lower income, *n* (%)	51 (24.6%)	22 (13.9%)	101 (11.8%)
	Lower middle income, *n* (%)	128 (61.8%)	93 (58.9%)	434 (50.8%)
	Upper middle income, *n* (%)	27 (13%)	43 (27.2%)	310 (36.3%)
	Higher income, *n* (%)	1 (0.5%)	0 (0%)	9 (1.1%)
Student or worker in the healthcare sector	No, *n* (%)	120 (58%)	105 (66.5%)	520 (60.9%)
	Yes, *n* (%)	87 (42%)	53 (33.5%)	334 (39.1%)
Source of information	Family, *n* (%)	3 (1.4%)	6 (3.8%)	18 (2.1%)
	Health workers, *n* (%)	122 (58.9%)	87 (55.1%)	556 (65.1%)
	None, *n* (%)	16 (7.7%)	10 (6.3%)	55 (6.4%)
	Radio/TV, *n* (%)	26 (12.6%)	20 (12.7%)	88 (10.3%)
	Social media, *n* (%)	27 (13%)	20 (12.7%)	72 (8.4%)
	Other, *n* (%)	13 (6.3%)	15 (9.5%)	65 (7.6%)
Presence of underlying disease	No, *n* (%)	183 (88.4%)	145 (91.8%)	730 (85.5%)
	Yes, *n* (%)	24 (11.6%)	13 (8.2%)	124 (14.5%)
Have you been tested for COVID-19?	Negative result	77 (37.2%)	66 (41.8%)	398 (46.6%)
	Positive result	17 (8.2%)	16 (10.1%)	96 (11.2%)
	Not tested	113 (54.6%)	76 (48.1%)	360 (42.2%)
How worried/fearful are you about getting infected or re-infected by the coronavirus?	Not at all concerned, *n* (%)	12 (5.8%)	11 (7.0%)	62 (7.3%)
	A little worried, *n* (%)	36 (17.4%)	24 (15.2%)	122 (14.3%)
	Moderately worried, *n* (%)	81 (39.1%)	47 (29.7%)	295 (34.5%)
	Very worried, *n* (%)	48 (23.2%)	53 (33.5%)	240 (28.1%)
	Extremely worried, *n* (%)	30 (14.5%)	23 (14.6%)	135 (15.8%)
Opinions about COVID-19 vaccines				
In your opinion, can COVID-19 infection be prevented with a vaccine?				
No, *n* (%)		87 (42%)	47 (29.8%)	249 (29.1%)
Yes, *n* (%)		120 (58%)	111 (70.3%)	605 (70.8%)
In your understanding, is there currently an effective vaccine against COVID-19?				
No, *n* (%)		110 (53.1%)	67 (42.5%)	348 (40.7%)
Yes, *n* (%)		97 (46.9%)	91 (57.6%)	506 (59.3%)
Can someone be re-infected with coronavirus after recovering from a previous COVID-19 infection?				
No, *n* (%)		30 (14.5%)	19 (12%)	99 (11.6%)
Yes, *n* (%)		177 (85.5%)	139 (88%)	755 (88.4%)

**Table 2 vaccines-09-00678-t002:** Factors associated with COVID-19 vaccine acceptance in Ecuador.

Covariates	Crude OR(95% CI)	Adjusted OR (95% CI) Full Model	*p*-Value
Age	1.05 (1.04–1.07)	1.04 (1.03–1.06)	<0.001
Gender			
Male	Ref	Ref	
Female	0.85 (0.67–1.08)	0.88 (0.67–1.15)	0.345
Education			
Secondary	Ref	Ref	
Undergraduate	1.62 (1.22–2.16)	1.34 (0.98–1.82)	0.067
Postgraduate	4.42 (3.13–6.25)	1.85 (1.17–2.93)	0.008
Student healthcare worker			
No	Ref	Ref	
Yes	1.19 (0.94–1.52)	1.22 (0.93–1.59)	0.151
Status of infection			
Negative	Ref	Ref	
Positive	1.80 (1.15–2.81)	2.13 (1.32–3.44)	0.001
Results not known	0.88 (0.69–1.13)	1.23 (0.93–1.62)	0.151
Level of worry about status			
Not worried	Ref	Ref	
A little worried	1.98 (1.17–3.33)	1.92 (1.08–3.40)	0.025
Moderately concerned	1.69 (1.06–2.70)	1.58 (0.94–2.64)	0.081
Very worried	3.15 (1.93–5.13)	3.02 (1.77–5.17)	<0.001
Extremely worried	3.26 (1.91–5.56)	3.40 (1.89–6.10)	<0.001
Presence of underlying disease			
No	Ref	Ref	
Yes	1.25 (0.87–1.78)	0.71 (0.48–1.06)	0.093
Can COVID-19 infection be prevented with a vaccine?			
No			
Yes	Ref	Ref	
	2.31 (1.80–2.97)	1.80 (1.33–2.44)	<0.001
In your understanding, is there currently an effective vaccine against COVID-19?			
No		Ref	
Yes	Ref	1.34 (1.12–1.79)	
	2.03 (1.60–2.57)		0.047
Area of residency			
Rural	Ref	Ref	
Suburban	1.08 (0.71–1.64)	0.87 (0.55–1.36)	0.529
Urban	1.76 (1.29–2.40)	1.16 (0.83–1.63)	0.379

## Data Availability

Data are available on the International Consortium (International Citizen Project COVID-19 (ICPCovid): http://www.icpcovid.com (accessed on 24 April 2021)) website and may be used by other investigators on request. De-identified participant data are available.

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
