# Peer review of "COVID-19 Vaccine Acceptance in Azuay Province, Ecuador: A Cross-Sectional Online Survey"

_vaccines, 2021, doi:10.3390/vaccines9060678_

Round 1

Reviewer 1 Report

This study addresses the important question of determinants and antecedents of vaccine uptake or hesitancy around COVID-19 vaccination. It focusses on Equator, which provides an interesting point to the international perspective of the epidemic vaccine response.

The manuscript is well written, I only have three essential points to comment on.

Why has gender not been included in the multivariate analysis?

Given the participant recruitment through snowballing the study cannot conclude on prevalences. The discussion should be modified accordingly. Alternatively, the analyses should take into account variables that could have been decisive for participation, for example by stratification.

A major limitation of the study comes from the limited to scope of the questionnaire, whose development was not based on current knowledge about vaccine hesitancy [see for example Betsch, PLoSONE 2018]. Confidence, collective benefits and calculation are not addressed, making this evaluation relatively incomplete. This should at least be recognised in the discussion.

Author Response

Response to reviewer 1

Point 1

This study addresses the important question of determinants and antecedents of vaccine uptake or hesitancy around COVID-19 vaccination. It focusses on Equator, which provides an interesting point to the international perspective of the epidemic vaccine response.

The manuscript is well written, I only have three essential points to comment on.

Why has gender not been included in the multivariate analysis?

Response 1

We now include gender in the multivariable analysis

Point 2

Given the participant recruitment through snowballing the study cannot conclude on prevalences. The discussion should be modified accordingly. Alternatively, the analyses should take into account variables that could have been decisive for participation, for example by stratification.

Response 2

Prevalences cited in our paper only refers to the prevalence of our study participants and therefore cannot be considered as prevalences of the general population. We mention this in the limitations of our study.

A large proportion 474/1219 (38 .9%) of our participants were health care workers or students working in the health sector. Even these health care workers cannot be considered as representative of all health care workers in the province where the survey was done. We now include the prevalence of vaccine acceptance among health care workers. 

Point 3

A major limitation of the study comes from the limited to scope of the questionnaire, whose development was not based on current knowledge about vaccine hesitancy [see for example Betsch, PLoSONE 2018]. Confidence, collective benefits and calculation are not addressed, making this evaluation relatively incomplete. This should at least be recognised in the discussion

Response 3

We know mention this as a limitation of the study. In the discussion we state “Our questionnaire additionally assessed most, but not all factors influencing vaccine hesitancy, because it did not consider all factors of the recently introduced 5C scale model. This model has been proposed, instead of the 3C scale model (confidence, complacency and convenience), to monitor vaccine hesitancy by assessing acceptance, access, affordability, awareness and activation.”

We also include the reference of Betsch et al.

Betsch C, Schmid P, Heinemeier D, Korn L, Holtmann C, Böhm R.Beyond confidence: Development of a measure assessing the 5C psychological antecedents of vaccination. PLoS One. 2018 Dec 7;13(12):e0208601

Moreover in the conclusion we state “Finally, follow up surveys to monitor vaccine hesitancy in random samples of populations should be considered using a validated scale for measuring vaccine hesitancy.”

Reviewer 2 Report

The manuscript described a survey on COVID-19 vaccine acceptance at a specific region in a specific country--Azuay Province of Ecuador. The results showed that there was a direct proportional of vaccine acceptance to the effectiveness of the vaccine, and factors such as age, level of education, “fear of contracting COVID-19”, and “understanding that a vaccine can prevent COVID-19” were associated with higher acceptance rate.  The major reason for vaccine hesitance was fear of side effect, followed by doubt about vaccine effectiveness.

Whereas this manuscript is well written and appears to have all the elements for a survey-type manuscript, and the results were in par with “common sense”, the deficiencies of this manuscript were identified by authors’ own admission in 5.1 (Limitations of the Study) such as sampling bias, etc.

However, a major flaw in this manuscript is the lack of “control questions” in the survey--the instrument was not vigorously tested and validated. Therefore, the scientific contribution of this manuscript is questionable.

Minor points: In Table 2, the column for P-values appears to be offset by a row. Also, for “status of infection”, it seems to contradict what was said in the Abstract, that is, a history of “negative COVID-19 test” rather than “positive”.

In short, the authors are advised to conduct further surveys by improving the instrument and by a better sampling strategy.

Author Response

Response to Reviewer 2

Point 1

The manuscript described a survey on COVID-19 vaccine acceptance at a specific region in a specific country--Azuay Province of Ecuador. The results showed that there was a direct proportional of vaccine acceptance to the effectiveness of the vaccine, and factors such as age, level of education, “fear of contracting COVID-19”, and “understanding that a vaccine can prevent COVID-19” were associated with higher acceptance rate.  The major reason for vaccine hesitance was fear of side effect, followed by doubt about vaccine effectiveness.

Whereas this manuscript is well written and appears to have all the elements for a survey-type manuscript, and the results were in par with “common sense”, the deficiencies of this manuscript were identified by authors’ own admission in 5.1 (Limitations of the Study) such as sampling bias, etc.

However, a major flaw in this manuscript is the lack of “control questions” in the survey--the instrument was not vigorously tested and validated. Therefore, the scientific contribution of this manuscript is questionable.

Response 1

We now mention as a limitation of the study the lack of “control questions” in the survey and that the instrument was not vigorously tested and validated. We state in the discussion “Third, we did not use a validated questionnaire that contained control questions. “

Point 2

Minor points: In Table 2, the column for P-values appears to be offset by a row. Also, for “status of infection”, it seems to contradict what was said in the Abstract, that is, a history of “negative COVID-19 test” rather than “positive”.

Response 2

Sorry for this. We corrected Table 2 and the abstract

Point 3

In short, the authors are advised to conduct further surveys by improving the instrument and by a better sampling strategy

Response 3

We now mention this in the conclusions. We state “Finally, follow up surveys to monitor vaccine hesitancy in random samples of populations should be considered using a validated scale for measuring vaccine hesitancy.”

Reviewer 3 Report

The authors did a lot of research with interesting results. However, there are some minor details to improve understanding of the article.
1) This study is multinational, so it would be important to delve more deeply into risk factors for non-vaccination derived from this study in other countries or from other vaccines in adults such as influenza, pneumococcus, MMR, etc.
2) In the tables of the results, the value of p is not possible to identify the correspondence with the variable due to the lack of lines or that it is slightly outside the row.
3) It is necessary to make a list, to describe all the questions that the study covers, it may be convenient to place it at least as annexes.
4) Table 1, the variable is explored by the geographical area where the population lives on the slope. The result variable is the acceptance of the vaccine in Ecuador, so the table does not provide adequate information in relation to the variable of interest of the study but to the area of ​​residence. This table needs to be changed.
5) For figures 1 and 2 it is necessary to calculate the confidence intervals of the reported percentages.
6) The multivariate logistic regression analysis could have a sensitivity analysis, perhaps stratified by area or by schooling, which are factors that have already been studied that speak about vaccines and could be used to propose information campaigns in relation to the main barriers to vaccination.

Author Response

R

Response to Reviewer 3

The authors did a lot of research with interesting results. However, there are some minor details to improve understanding of the article.

Point 1

This study is multinational, so it would be important to delve more deeply into risk factors for non-vaccination derived from this study in other countries or from other vaccines in adults such as influenza, pneumococcus, MMR, etc.

Response 1

We now included in the discussion: “In a recent survey about COVID-19 acceptance with the same ICPcovid questionnaire conducted in several other low and middle income countries, the vaccine acceptance rate in Brazil was 94.2%, in Thailand 87.3%, and in Malaysia 78.6%, but much lower in most African countries, for example only 59.4% in the DRC for a vaccine that is 95% effective the COVID-19.”

Point 2

In the tables of the results, the value of p is not possible to identify the correspondence with the variable due to the lack of lines or that it is slightly outside the row.

Response 2

Sorry for this. We now corrected the Table

Point 3

It is necessary to make a list, to describe all the questions that the study covers, it may be convenient to place it at least as annexes.

Response 3

We now included the questionnaire as supplementary information

Point 4

Table 1, the variable is explored by the geographical area where the population lives on the slope. The result variable is the acceptance of the vaccine in Ecuador, so the table does not provide adequate information in relation to the variable of interest of the study but to the area of ​​residence. This table needs to be changed.

Response 4

Point 5

For figures 1 and 2 it is necessary to calculate the confidence intervals of the reported percentages.

Response 5

We now included the confidence intervals

Point 6

The multivariate logistic regression analysis could have a sensitivity analysis, perhaps stratified by area or by schooling, which are factors that have already been studied that speak about vaccines and could be used to propose information campaigns in relation to the main barriers to vaccination

Response 6

We now performed a sensitivity analysis stratified by level of schooling

We considered area of residence as a random effect and fitted a generalised linear mixed effects model to account for cluster effect of area of residence. Indeed there were not much changes to the results, moreover the estimate for the variability accounted by the level of area was 0.004359 with a standard deviation of 0.06602. We carried out similar analyses considering level of schooling as a random effect to account for cluster effect of the level of education, yet again the model was robust and the variability accounted for by level of schooling is 0.05393 and standard deviation 0.2322 . We conclude that, after accounting for the effect of area of residence and schooling as clusters, we did not observe significant changes in the results of our multiple logistic regression analysis. Therefore we propose not to include the details of this analysis in the paper.

We now only mention in the text: “In a sensitivity analysis accounting for the effect of area of residence and schooling as clusters, we did not observe significant changes in the results of our multiple logistic regression analysis (details not shown).”

Round 2

Reviewer 2 Report

A scientific survey should have vigorous sampling strategy with validated instrument composed of internal controls. Furthermore, it is difficult to judge how the layperson in the surveyed population really understand vaccine effectiveness--50%, 70%, or 90%--to make an informed decision. 

Although stated by the author's responses, the survey questions were not attached nor found. 

However, given the results were not significantly departed from "common sense", and with the interests in this topic, plus the authors' explicitly stated limitations, this reviewer is reluctantly not objecting to publication of this manuscript.

Author Response

Response to reviewer

Comment reviewer

A scientific survey should have vigorous sampling strategy with validated instrument composed of internal controls. Furthermore, it is difficult to judge how the layperson in the surveyed population really understand vaccine effectiveness--50%, 70%, or 90%--to make an informed decision. 

Although stated by the author's responses, the survey questions were not attached nor found. 

However, given the results were not significantly departed from "common sense", and with the interests in this topic, plus the authors' explicitly stated limitations, this reviewer is reluctantly not objecting to publication of this manuscript

Response

We agree completely with the reviewer that scientific surveys should have a vigorous sampling strategy with a validated instrument composed of internal controls. However we believe it is also important, when confronted with an emergency such as the COVID-19 pandemic, to consider less rigorous methodologies to obtain rapid information to plan public health interventions. Information concerning COVID-19 vaccine willingness is important to obtain ideally before a vaccination campaign is started. We have seen in several low income countries that COVID-19 vaccines were introduced without information about COVID-19 vaccine willingness and that vaccines were lost because governments and the population were not prepared for it.

Our findings should be a starting point for more in depth research about COVID-19 vaccine willingness and hesitancy. In the conclusion of our paper we state

“follow up surveys to monitor vaccine hesitancy in random samples of populations should be considered using a validated scale for measuring vaccine hesitancy.”

We hope that in its present form our paper is publishable in vaccines.

We thank the reviewer not to object to the publication of our paper.